# PET Imaging of the Neuropeptide Y System: A Systematic Review

**DOI:** 10.3390/molecules27123726

**Published:** 2022-06-09

**Authors:** Inês C. F. Fonseca, Miguel Castelo-Branco, Cláudia Cavadas, Antero J. Abrunhosa

**Affiliations:** 1CIBIT/ICNAS—Institute for Nuclear Sciences Applied to Health, University of Coimbra, 3000-548 Coimbra, Portugal; inesfonseca@icnas.uc.pt (I.C.F.F.); mcbranco@fmed.uc.pt (M.C.-B.); 2Faculty of Pharmacy, University of Coimbra, 3000-548 Coimbra, Portugal; ccavadas@ci.uc.pt; 3ICNAS Pharma Unipessoal, Lda, Ed. ICNAS, Pólo das Ciências da Saúde, University of Coimbra, 3000-548 Coimbra, Portugal; 4Faculty of Medicine, University of Coimbra, 3000-548 Coimbra, Portugal; 5CNC—Center for Neuroscience and Cell Biology, University of Coimbra, 3004-504 Coimbra, Portugal; 6CIBB—Centre for Innovative Biomedicine and Biotechnology, University of Coimbra, 3004-531 Coimbra, Portugal

**Keywords:** neuropeptide Y, positron emission tomography, neuroimaging, PET radiotracers, NPY receptors

## Abstract

Neuropeptide Y (NPY) is a vastly studied biological peptide with numerous physiological functions that activate the NPY receptor family (Y_1_, Y_2_, Y_4_ and Y_5_). Moreover, these receptors are correlated with the pathophysiology of several diseases such as feeding disorders, anxiety, metabolic diseases, neurodegenerative diseases, some types of cancers and others. In order to deepen the knowledge of NPY receptors’ functions and molecular mechanisms, neuroimaging techniques such as positron emission tomography (PET) have been used. The development of new radiotracers for the different NPY receptors and their subsequent PET studies have led to significant insights into molecular mechanisms involving NPY receptors. This article provides a systematic review of the imaging biomarkers that have been developed as PET tracers in order to study the NPY receptor family.

## 1. Introduction

Molecular imaging has been of immense value to the field of brain research, as the inaccessibility of the brain represents one of the greatest obstacles to central nervous system (CNS) studies [1]. Non-invasive imaging technology has not only played a major role in the discovery and development of CNS drugs but it has also become a prominent approach to the study and clarification of diverse neurological pathways. The uniqueness of the CNS field lies in the existence of its many potential drug targets; however, only a few are being exploited by current CNS therapeutics [2]. CNS diseases are highly heterogenous, which complicates the elucidation and research of their physiology and can lead to failure in the development of a new potential drug.

The potential of using imaging to optimize CNS drugs’ discovery and development relies on various imaging techniques such as computerized tomography (CT), magnetic resonance imaging (MRI), single photon emission computed tomography (SPECT), positron emission tomography (PET) and others [3,4,5]. The main characteristics of these imaging techniques are summarized in Table 1.

The PET imaging technique uses the distinct decay characteristics of low-dose, high specific activity, short half-life positron-emitting radionuclides. These radionuclides (e.g., ^11^C, ^13^N, ^18^F, ^68^Ga and ^64^Cu), which are usually produced in a cyclotron, are involved in the design and production of radiolabeled tracers that are expected to interact selectively with the target of interest, allowing the in vivo tracing of biochemical processes. PET is an extremely sensitive technique as it allows radioligands to be detected at picomolar to femtomolar levels. Concerning this and the fact that the positron emitting radioisotopes decay with a reasonably short half-life (e.g., 20 min for ^11^C and 110 min for ^18^F), it becomes possible to administer high doses so as to provide a strong imaging signal without producing the long-term health risks that are related to ionizing radiation. Hence, PET allows the quantitative mapping of the spatial distribution of the radiolabeled tracers in living animal species and in living humans [2,3,4,5,6,7,8,9,10,11,12]. Hereby, brain positron emission tomography has led to new findings and changes to the standard concepts in neuroscience; namely in cerebrovascular diseases, movement disorders, dementia, epilepsy, schizophrenia, addiction, depression, anxiety and cerebral tumors [13].

Neuropeptide Y (NPY) is one of the most abundant neuropeptides in the mammalian brain. It is known to participate in essential physiological functions in humans and has been correlated with the pathophysiology of several diseases such as feeding disorders, anxiety, metabolic diseases, neurodegenerative diseases, several types of cancer and others [14,15,16,17,18,19]. However, despite all of the research around this topic, the neurological pathways that are related to NPY are not quite clearly understood yet. As such, PET imaging emerges as a powerful technique that may be used to study the neuromolecular mechanisms of this neuropeptide.

## 2. Methodology

The present research was conducted through the utilization of the PubMed, Web of Science and Google Scholar databases (accessed between March 2020 and February 2022), using the following both as text and as MeSH terms: “Neuropeptide Y”, “NPY”, “NPY receptors”, “PET”, “PET imaging” and “NPY PET imaging”. Only articles in English were reviewed. The systematic literature research process included a total of 194 articles. According to the PRISMA flowchart, after duplicate removal, 113 articles have been considered, fully read, analyzed and extensively described according to their title and abstract [20]. Other relevant articles were also checked after they had been found in the references of the articles that were originally included in the retrieved literature.

## 3. Overview of Neuropeptide Y Family and Receptors

Neuropeptide Y (NPY) is a remarkably conserved 36-amino-acid peptide, being one of the most abundant peptides in the central nervous system of mammals [21,22]. NPY is structurally similar to peptide YY (PYY) and pancreatic peptide (PP) [23]. These three peptides constitute a family of regulatory peptides and they share a common hairpin-like three-dimensional structure, which is known as the PP-fold [24]. However, they differ in their main location as NPY is primarily synthesized and released by neurons, PP is mainly found in endocrine pancreas and PYY is expressed in both neurons and gut endocrine cells. PYY is essentially produced in gut endocrine cells and released postprandially, a process which inhibits pancreatic secretion and gut motility [25,26]. PYY also reduces appetite [27]. Similarly, PP reduces pancreatic secretion, gut motility and appetite as well [28,29]. NPY is widely distributed in the central nervous system (CNS) as it is present in the basal ganglia, limbic system, amygdala, hippocampus, hypothalamus, septal nuclei and nucleus accumbens [30,31]. NPY is also detected in the peripheral nervous system, namely in the sympathetic nervous system, where it is co-localized with norepinephrine, and in a subpopulation of parasympathetic neurons [32].

NPY’s physiological processes are mediated through the activation of different subtypes of NPY receptors belonging to the G-protein coupled receptors (GPCR) super family. The NPY receptors act via pertussis toxin-sensitive G-proteins, i.e., members of the G_i_ and G_o_ family, leading to the inhibition of adenylyl cyclase and, consequently, the decrease of cAMP accumulation in tissues and cells [33]. To date, six subtypes of NPY receptors have been described: Y_1_, Y_2_, Y_3_, Y_4_, Y_5_ and y_6_. Until now, five of these have been cloned; with the exception of the Y_3_ receptor, which has not been cloned and no specific agonist or antagonist have been attributed to it [34]. Comparisons between their sequences demonstrate that Y_1_, Y_4_ and y_6_ are more similar to each other than they are to the receptors Y_2_ and Y_5_. Concerning the y_6_ receptor, it has been cloned though its function is undetermined because it encodes for a truncated non-functional receptor in humans [33].

The Y_1_ receptor was the first NPY receptor to be cloned. It embodies 384 amino acids and is typically expressed in postsynaptic sites. Its main agonists are NPY and PYY, even though it can also be activated by PP with a minor level of potency (Table 2). Highly selective agonists and antagonists with high binding affinities to the receptor subtype have been developed, shedding light on its physiological effects. Regarding Y_1_ antagonists, the first non-peptide antagonist, which was highly potent and selective, that was discovered was (*R*)-N^2^-(diphenylacetyl)-*N*-[(4-hydroxyphenyl)methyl]-argininamide, commonly known as BIBP3226, that displays its affinity in the nanomolar range [35]. The high selectivity and specific binding to the Y_1_ receptor has been demonstrated in numerous binding assays and in vitro and in vivo bioassays, with BIBP3226 being the most characterized Y_1_ receptor antagonist.

The Y_2_ receptor is a 381 amino acid protein with high affinities to NPY and PYY (Table 2). The particularity of this receptor lies in its presynaptic localization, leading to an autoreceptor role as it is involved in the inhibition of the release of NPY and other neurotransmitters [36]. In comparison to the Y_1_ receptor, lower densities of expression of this receptor are observed in the mammalian brain [37].

The Y_4_ receptor, a 375 amino acid peptide, is the only member of the NPY receptor family which has great affinity with PP as a ligand. PYY and NPY can also activate the receptor but they do so at a lower level of affinity (Table 2). The development of a peptide Y_4_ agonist, BVD-74D, showed that it exhibited picomolar affinity for the receptor and it suppressed the food and water intake and weight gain of normal mice that were fed with normal diets and the food intake of normal mice that were fed with high-fat diets [38,39].

Concerning the Y_5_ receptor, it has two isoforms that are pharmacologically alike: one long one that embodies 455 amino acids and one short one that is a splice variant that lacks the first 10 amino acids [40]. Both of these isoforms bind efficiently to NPY and PYY and have a lower affinity with PP (Table 2). The Y_5_ receptor is strongly correlated with food intake, having been named initially as the “feeding receptor”. Besides its localization being in several areas of the brain—mainly the ones that regulate food intake, such as the hypothalamus—it was reported that NPY stimulated feeding via the Y_5_ receptor [41]. Erondu et al. reported their results concerning preclinical and clinical studies with the aim of testing the hypothesis that the antagonism of the Y_5_ receptor leads to weight loss in overweight and obese patients [42]. The antagonist that was used in these studies was MK-0557 [43], a potent, highly selective, orally active Y_5_ antagonist and the PET ligand was [^11^C]MK0233, which has been used for receptor occupancy studies in rhesus monkeys and human volunteers. Interestingly, the authors concluded that Y_5_ receptor antagonism did not induce clinically meaningful weight loss in obese patients.

**Table 2 molecules-27-03726-t002:** Characterization of NPY receptor subtypes.

Receptors	Endogenous Agonists in Order of Potency	Selective Agonists	Selective Antagonists	Expression	Functions
Y_1_	NPY (0.2 nM) ≥ PYY (0.7 nM) >> PP (>100 nM) [33]	[Leu^31^,Pro^34^]NPY [44]; [Pro^34^]NPY[Leu^31^,Pro^34^]PYY [45]; [Pro^34^]PYY; [Phe^7^,Pro^34^]NPY [46]	BIBP3226 [35]; BIBO3304 [47]; SR120819A [48]; LY357897 [49]; J-115814 [50]; GI264879A [51]; GW1229 [52]; H394/84 [53]; PD160170 [54]; BMS-193885 [55]; BMS-205749 [56]; J-104870 [57]; BW1911U90[58]; T-190 [59]; T-241 [60]	Cerebral cortex, hippocampus, thalamus, hypothalamus, vascular smooth muscle cells, adipose tissue, kidney and gastrointestinal tract [61]	Feeding behavior [62,63]; inhibition of nociception [64]; regulation of hormone secretion [65]; ethanol consumption [66,67,68]; emotional behavior [69] and stress response [70]
Y_2_	NPY (0.7 nM) ≈ PYY (0.7 nM) >> PP (>1000 nM) [33]	NPY_3-36_; NPY_13-36_; PYY_3-36_, PYY_13-36_ [45]; TASP-V [71]; Ac-[Lys^28^, Glu^32^]- (25–36)-NPY [72]	T_4_[NPY_33-36_]_4_ [73]; BIIE0246 [74]; JNJ-5207787 [75]; JNJ-31020028 [76]; SF-11, SF-21, SF-22; SF-31, SF-41 [77]; ML072-ML075 [78]; CYM 948 and CYM 9552 [79]; 36 (GSK) [80]; 2 (GSK) [81]; [^3^H]UR-PLN196 [82]	Hippocampus, hypothalamus, thalamus, amygdala, brainstem, spleen, liver, blood vessels, gastrointestinal tract and fat tissue [83,84,85]	Feeding behavior, anxiety, neuronal excitability and epilepsy and angiogenesis [86]; circadian rhythm [87]; alcohol dependence [88]; cognitive processes [89] and locomotor activity [90]
Y_3_	NPY ≥ NPY_13-36_ >> PYY, PP [18]	__	__	Hippocampus [91]; brain stem [92]; human adrenal medulla, rat superior cervical ganglia sympathetic neurons, rat *nucleus tractus solitarius*, rat cardiac ventricular membranes and rat distal colon [34,93]	Mediation of the NPY-induced secretion of catecholamines [93]
Y_4_	PP (0.05 nM) > NPY ≈ PYY [33]	PP; GW1229 [94]; BW1911U90, T-190, T-241 [95]; BVD-74D [38];	UR-AK49 [96]	Hypothalamus, cerebral cortex, colon, small intestine, prostate, pancreas, skeletal muscle, thyroid gland, heart, stomach, adrenal medulla and nasal mucosa [97]	Food intake regulation [98]; luteinizing hormone release [99] and neuroprotection [100]
Y_5_	NPY (0.6 nM) ≥ PYY (1 nM) ≥ PP [33]	D-[Trp^32^]NPY [41]; D-[Trp^34^]NPY [101]; [Ala^31^,Aib^32^]NPY [102]; [cPP^1–7^, NPY^19–23^, Ala^31^, Aib^32^,Gln^34^]-hPP [103]; [^125^I][hPP^1–17^,Ala^31^, Aib^32^]NPY [104]; [^125^I][cPP^1–7^, NPY^19–23^, Ala^31^, Aib^32^,Gln^34^]-hPP [105]; BWX-46 [106]	CGP71683A [107]; L-152,804 [108]; FMS-586 [109]; Lu AA33810 [110]; S 25585 [111]; NPY5RA-972 [112]; FR240662, FR252384 [113]; GW438014A [114]; NTNCB [115]; S-2367 (Velneperit) [116]; [^11^C]MK-0233 [42]; MK-0557 [43]; [^35^S]SCH 500946 [117]	Hypothalamus, cerebral cortex, amygdala, hippocampus and *Substantia nigra* [118] and intestine, ovary, testis, prostate, spleen, pancreas, kidney, skeletal muscle, liver, placenta and heart [97]	Appetite regulation, anxiolytic and anticonvulsant, regulation of circadian rhythms and inhibition of LH release [97]
y_6_	NPY ≈ PYY > PP [33]	__	__	Hippocampus, hypothalamus, heart and skeletal muscle [119,120]	Promotion of lean and bone mass acquisition in mice [121]

## 4. Development of PET Tracers for NPY Receptors

As PET imaging is a powerful tool for drug development, especially when the target is in the central nervous system, the ability to develop successful radiotracers becomes significant as well. Radiotracers can have different roles in the drug development process: they can be used to demonstrate target delivery, to observe the change in the biochemistry of a system and to show the occupancy of a targeted site. In order to obtain an excellent radiotracer, some requirements must be fulfilled. Firstly, the radiotracer needs to have a high level of affinity towards the target, which requires the prior identification of the target of interest and then the development of ligands with high affinity to the target. Ideally, it is stipulated that the ratio B_max_/*K*_d_ > 10 (as B_max_ represents the target site concentration and *K*_d_ the equilibrium dissociation constant of the radiotracer for that site) is met in order to have a high probability of the detection of the specific binding in vivo (even though there are some exceptions). Lipophilicity, expressed as Log P, is of relevance as it can influence the extent of the nonspecific binding activity and the ability to cross lipid membranes. Log P should be between 1 and 3.5 (though it is preferable that it is less than 3) for appropriate brain penetration and a good specific–non-specific binding ratio. In addition, the radiotracer must not be a P-glycoprotein (P-gp) substrate, as this is an efflux transporter that is substantially expressed in blood–brain barrier (BBB) penetration, blocking brain penetration. Ultimately, the radioligand should have an easy radiolabeling precursor, which means that the ligand should have chemical structural features that are suitable for radiolabeling with the appropriate radionuclide. The main goal in this step is to utilize a radiotracer with high molar activity, high radiochemical purity and high radiochemical yield. For further information about the main aspects of in vivo site-directed radiotracers, see Patel et al. [122].

### 4.1. NPY Y_1_ Receptor

Over the past years, Y_1_ receptor antagonists have been developed in order to study the diverse physiological roles of their receptors. In order to further understand the physiological role of Y_1_ receptors in vivo, the development of novel PET tracers that are suitable for Y_1_ receptors would allow the non-invasive imaging of the same and the determination of their occupancy. Bearing this in mind, Kameda et al. synthesized a series of 2,4-diaminopyridines and evaluated their suitability to become a PET tracer for the Y_1_ receptor [123]. This investigation started with the identification of lead compound **1** (Figure 1) as a promising PET tracer during structure-activity relationship studies. Compound **1** showed appropriate lipophilicity (Log P = 3.0) and favorable characteristics to be radiolabeled with ^18^F; however, it was shown to have only a moderate level of Y_1_ receptor binding affinity (Y_1_ IC_50_ = 21 nM). Thus, the focus of the study was on improving the affinity of the lead compound **1** towards the Y_1_ receptor and its lipophilicity.

The SAR studies led to the identification of the potent and selective compounds **1d** and **1e** (Table 3 and Table 4) as promising candidates for Y_1_ PET tracers. Firstly, the variation of the antagonist **1** right hand 2-amino group was examined and the 2-fluoro-6-methylenepyridine derivative showed the most potent Y_1_ affinity (Y_1_ IC_50_ = 1.5 nM, compound **1a**, Table 3). Bearing this result, this derivative was substituted with other proper functional groups in order to favor its use in radiolabeling. The replacement of the fluorine with a methyl group resulted in a powerful increase in the level of binding affinity (Y_1_ IC_50_ = 0.69 nM, compound **1b**, Table 3) and a subtle reduction of the lipophilicity. Then, the left hand heterocycle portion of the previous compound was optimized. The 4,5-dimethyloxazole ring was replaced with a 4,5-dimethylthiazole ring, an atom of fluorine was attached to the 5-methyl group (compound **1c**, Table 3) and, lastly, the fluorine was moved from the 5-methyl to a 4-methyl group of the thiazole ring so as to produce compound **1d** with its high affinity towards the Y_1_ receptor and reduced lipophilicity (Table 3). Notably, this fluorine substitution is an additional labelling option for the incorporation of ^18^F via nucleophilic substitution. Moreover, the 4-fluoromethyl-5-ethyl-thiazole derivative (compound **1e**) showed an even further improvement in Y_1_ receptor binding and reasonable lipophilicity (Table 3). Both compounds **1d** and **1e** also exhibited great selectivity towards the Y_1_ receptor (Table 4).

Afterwards, Hostetler et al. identified Y_1_-973 (compound **1e**) as a promising PET tracer candidate to be radiolabeled as it fulfilled the abovementioned requirements: potency < 0.5 nM, Log P < 3.5, P-gp ratio < 3 and it was suitable for radiolabeling with ^18^F. Thus, the authors were able to report the synthesis and pre-clinical evaluation of the NPY Y_1_ receptor PET tracer [^18^F]Y_1_-973 [124]. [^18^F]Y_1_-973 was produced through the reaction of [^18^F]KF/K_222_ with the t-butyloxycarbonyl (Boc)-protected chloromethyl thiazole **2** precursor in DMSO at 130 °C using microwave heating, followed by deprotection with 1 M HCl (Figure 2). [^18^F]Y_1_-973 was obtained in satisfactory radiochemical yields (18 ± 13%, *n* = 9) with >98% radiochemical purity and high molar activity (1548 ± 859 Ci/mmol), in a total synthesis time, including purification by semi-preparative HPLC, of about 45 min [124].

In vivo PET imaging studies of [^18^F]Y_1_-973 in rhesus monkeys have demonstrated that this PET tracer promptly penetrates the blood–brain barrier. Baseline PET scans of this radiotracer in rhesus monkeys have shown heterogeneous distribution throughout the brain, with a pattern of uptake that is consistent with the known high NPY Y_1_ brain density (the highest level of uptake was in the striatum and cortical regions, there was moderate uptake in the thalamus and low uptake in the cerebellum). [^18^F]Y_1_-973 also exhibited rapid kinetics in rhesus monkeys’ brains, with the uptake peaking in the striatum at approximately 30 min. Additionally, it has a large binding potential that is suitable for receptor occupancy PET studies with an NPY Y_1_ antagonist [124].

Hofmann et al. reported a suitable 1^8^F-labeled high-molecular weight glycopeptide for the imaging of peripheral NPY Y_1_ receptor-positive tumors, namely breast cancer. Notably, the human Y_1_ receptor subtype has been discovered to be overly expressed in 85% of primary breast cancer and in 100% of lymph node metastases [125]. Despite [^18^F]Y_1_-973 having remarkable properties for brain imaging of the Y_1_ receptor, its high lipophilicity and significant liver accumulation makes it less suitable for peripheral application for breast cancer imaging. Therefore, the authors reported the development of a full length NPY analogue bearing a glycosylation site in position 4 of the amino acid sequence and the ^18^F-labeling by ^18^F-fluoroglycosylation using click chemistry (Figure 3) [126].

The authors selected [F^7^, P^34^] NPY analogue as it had scientific evidence supporting its selective binding to Y_1_ receptor and they then developed two peptide precursors bearing an alkyne-bearing functionality (peptides **5a** and **6a**, Figure 3), allowing regiospecific ^18^F-labeling by ^18^F-fluoroglycosylation using click chemistry. The alkyne functionalization was obtained by the introduction of propargylglycine (Pra) at position 4 (Figure 3).

^18^F-Fluoroglycosylated NPY analogues **[^18^F]5b** and **[^18^F]6b** were obtained through the course of different reactions (Figure 3). Firstly, the precursor 3,4,6-tri-*O*-acetyl-2-*O*-trifluoromethanesulfonyl-*β*-D-mannopyranosyl azide **3** reacted with [^18^F]fluoride, K_222_ and potassium carbonate in acetonitrile, resulting in the formation of 2-deoxy-2-[^18^F]fluoroglucopyranosyl azide **4**. Then, the remaining solution (60 mM NaOH) was adjusted to pH 8 (1M HCl) and a solution of the peptides **5a** or **6a**, sodium ascorbate, CuSO_4_ and THPTA (tris(hydroxypropyltriazolylmethyl)amine) were added. After isolation by semipreparative HPLC and subsequent solid phase extraction, **[^18^F]5b** (*n* = 7) or **[^18^F]6b** (*n* = 2) were obtained. The molar activities were about 40–70 GBq/μmol in overall radiochemical yields of 20–25% with a total synthesis time of 75 min. Further in vitro characterization was undertaken and the glycopeptide **[^18^F]5b** showed great selectivity for Y_1_ receptor over the Y_2_ receptor and strong Y_1_ receptor internalization. In this study, the preclinical animal model that was used was an MCF-7 breast cancer tumor model. PET imaging studies in these animals demonstrated a specific binding uptake of the glycopeptide **[^18^F]5b** from Y_1_ receptors present in MCF-7 tumors and that **[^18^F]5b** exhibited considerably increased renal clearance properties, compared to the NPY DOTA-derivatives, showing favorable kinetics in vivo [126].

As mentioned earlier, 85% of mammary carcinomas are Y_1_ receptor positive. To address this issue, Keller et al. reported the exploration of ^18^F-labeled compounds that were derived from the argininamide BIBP3326, which is a Y_1_ receptor antagonist (Figure 4) [127].

The authors acknowledged that the bioisosteric substitution of the guanidine group in the argininamide moiety of a BIBP3226 compound (e.g., UR-MK114, **7b** and UR-MK136, **7c**, Figure 4) was shown to lead to the formation of highly selective and potent fluorescent ligands, bivalent ligands and tritiated radioligands for the Y_1_ receptor. Similarly, Keller et al. used this strategy to design new and selective [^18^F] Y_1_ receptor PET ligands. After the synthesis and in vitro characterization of a series of fluorinated argininamides, they concluded that derivative **10** (Figure 5) was the most promising candidate as it displayed the lowest lipophilicity (Log P of 3.4). Derivative **10** was synthesized using the building block **8**. The radiosynthesis of **[^18^F]10** was proceeded by the treatment of excess of amine **9** with **[^18^F]8** (Figure 5). **[^18^F]8** was, beforehand, obtained by the ^18^F-fluorination of the 9′-anthrylmethyl-2-bromopropionate precursor, under mild conditions and in the presence of KH_2_PO_4_, followed by isolation by semipreparative HPLC. **[^18^F]10** was obtained in an overall RCY of 5–8% (uncorrected for decay) in an overall synthesis time of 70–80 min, with a radiochemical purity of >99% and a molar radioactivity of 12–21 GBq/μmol (*n* = 6) [127].

The tracer **[^18^F]10** was then evaluated in vitro and in vivo using tumor-bearing nude mice with Y_1_R positive MCF-7 tumors. Its Log P was experimentally determined resulting in a value of 2.34 ± 0.03. **[^18^F]10** had fast blood clearance time at 30 and 90 min, low uptake in the liver and high uptake in the gallbladder and intestines. The tracer also showed high tumor retention and a suitable signal-to-noise ratio for PET imaging. In comparison with **[^18^F]5b**, **[^18^F]10** had lower tumor uptake because of its uptake in the gallbladder. Nonetheless, **[^18^F]10** showed diminished kidney uptake when compared with the peptide tracer, which is the major advantage of the nonpeptide radioligand over the peptide tracer for PET imaging. In vivo PET imaging studies with the antagonist **[^18^F]10** in the MCF-7 nude mice revealed the visualization of Y_1_R-positive MCF-7 tumors [127]. 

Regarding these results and in order to reduce the lipophilicity of a candidate ligand and to achieve more suitable biodistribution with minimal biliary excretion, hence improving the tumor visibility in PET images, Maschauer et al. recently reported the synthesis and radiosynthesis of three BIBP3226 derivatives that were conjugated with ^18^F-fluoroethoxy linkers and ^18^F-fluoroglucosyl moiety and further in vitro and in vivo characterization [128]. **[^18^F]15** was obtained through a copper-catalyzed azide–alkyne cycloaddition (CuAAc-Based) ^18^F-fluoroglycosylation method between the BIBP3226 derived alkyne **11** and 6-deoxy-6[^18^F]fluoroglucosyl azide **[^18^F]12** (Figure 6). **[^18^F]12** was previously prepared and isolated by semi-preparative HPLC and deacetylated with NaOH, following a priorly described method [129]. **[^18^F]15** was then isolated by semi-preparative radio-HPLC, with a radioactivity yield (RAY) of about 20% (in reference to [^18^F]fluoride), molar activity of 9 GBq/μmol and radiochemical purity >99% in a total synthesis time of 80 min. **[^18^F]16** and **[^18^F]17** were attained using the same protocol as **[^18^F]15**, apart from the deprotection step. **[^18^F]13** and **[^18^F]14** were provided by the [^18^F]-labelling of their corresponding tosylate-bearing precursors, with radioactivity yields of 38% and 40%, in total synthesis times of 40 min and 35 min, respectively. Afterwards, they were treated with the BIBP3226-derived alkyne **11** in a click chemistry reaction under similar conditions as those which were used for the glucosyl derivative **[^18^F]15**. **[^18^F]16** and **[^18^F]17** were isolated with radioactivity yields of about 5% and 10% (in reference to [^18^F]fluoride), molar activities between 5–6 GBq/μmol and radiochemical purity >99% in a 60 min total synthesis time (Figure 6).

The in vitro characterization of the three compounds showed that these were selective towards the Y_1_ receptor and revealed their weak lipophilicity, with **[^18^F]15** being the most hydrophilic compound and **[^18^F]16** being the least hydrophilic one (**[^18^F]15** Log P = 0.78, calculated: 0.43; **[^18^F]17** Log P = 1.49, calculated: 1.94; **[^18^F]16** Log P = 1.74, calculated: 2.03). This characterization also showed the stability of all three of the radioligands as no radioactive degradation products were shown with 3 h of incubation. Regarding the cellular accumulation, the human breast cancer cells MCF-7-Y1 were used, wherein it was noticed that there was the highest specific accumulation of **[^18^F]16**, mainly explained by the high affinity of this ligand. The other two showed low or non-specific binding to MCF-7-Y1 cells. In autoradiography experiments, **[^18^F]16** and **[^18^F]17** have demonstrated marked specific binding to the solid MCF-7-Y1 tumors, whereas **[^18^F]15** showed very low specific binding. Biodistribution studies have indicated that, generally, all three of these radioligands are similar to **[^18^F]10** as moderate amounts of radioactivity were detected in the kidneys and intestines and extraordinarily high radioactivity was observed in the gallbladder. Thus, the aim of achieving a reduced level of uptake in the gallbladder by using more hydrophilic compounds was not reached, although it was possible to reduce the lipophilicity of the analogues. Blood sample analysis by HPLC revealed that the radiotracers **[^18^F]16** and **[^18^F]17** underwent rapid degradation, resulting in the formation of hydrophilic radiometabolites, while **[^18^F]15** was more stable in the blood in vivo. Finally, PET imaging was performed and the radiotracer **[^18^F]15** did not allow the visualization of the tumor, most likely due to its poor affinity to the receptor. Both radioligands **[^18^F]16** and **[^18^F]17** showed specific tumor accumulation in vivo, even though there were relatively high background values in the PET images, due to the formation of radiometabolites in the blood.

Recently, the radiolabeling of the Y_1_ antagonist BMS-193885 with [^11^C] has also been reported [130]. BMS-193885 (1,4-dihydro-[3-[[[[3-[4-(3-methoxyphenyl)-1-piperidinyl]propyl]amino]carbonyl]amino]phenyl]-2,6-dimethyl-3,5-pyridinedicarboxylic acid, dimethyl ester, **18**, Figure 7) is a potent and selective NPY Y_1_ receptor antagonist with good brain penetration and systemic bioavailability but poor oral bioavailability [55,131]. It was previously disclosed that it reduces food intake and body weight in animal models of obesity, being pharmacologically efficacious in the treatment of obesity in animal models [131]. Therefore, Kawamura et al. reported the radiolabeling of **18** in two different ways: the first one involved the methylation of the desmethyl analog **19** (via **A**, Figure 8) with [^11^C]methyl iodide and the other one involved the radiosynthesis of [^11^C]desmethyl BMS-193885 (**[^11^C]22**, via **B**, Figure 8), using [^11^C]phosgene as the radiolabeling agent [130].

In order to obtain the tracer **[^11^C]18**, an automated synthesis system that was developed in-house was used. A solution of **19** and sodium hydroxide solution in *N*,*N*-dimethylformamide (DMF) was added to a dry septum-equipped vial prior to the reaction. [^11^C]Methyl iodide was produced by the reduction of cyclotron-produced [^11^C]carbon dioxide with lithium aluminum hydride, followed by iodination with hydroiodic acid. The [^11^C]methyl iodide that was produced was then trapped in a reaction vial containing **19** in DMF. The reaction mixture was heated and held at 80 °C for 5 min. Afterwards, the solution was purified by preparative HPLC. **[^11^C]18** was synthesized approximately 30 min after the end of irradiation (EOI), with a radiochemical yield of 23 ± 3.2% (*n* = 6), molar activity of 87 ± 28 GBq/μmol (*n* = 6) and radiochemical purity >99% (Figure 8A). The radiosynthesis of **[^11^C]22** (Figure 8B) started with the initial production of [^11^C]carbon dioxide using a cyclotron, followed by the subsequent reduction of [^11^C]carbon dioxide with H_2_ in the presence catalyst, generating [^11^C]methane, the chlorination of [^11^C]methane and the conversion of [^11^C]carbon tetrachloride to [^11^C]phosgene. Afterwards, aniline derivative **20** in a tetrahydrofuran (THF) solution was added to a dry septum-equipped vial just prior to radiosynthesis. [^11^C]phosgene was trapped in the solution containing **20** at −15 °C for 1 min. The reaction mixture was heated at 30 °C for 3 min. When the solution cooled, it was added a solution containing amine derivative **21** in THF. Then, the reaction mixture was heated at 80 °C for 3 min in order to remove the THF. The crude product was then purified through the use of preparative HPLC. Finally, **[^11^C]22** was obtained at about 33 min after EOI with a radiochemical yield of 24 ± 1.5% (*n* = 4), molar activity of 52 ± 10 GBq/μmol (*n* = 4) and radiochemical purity >99% (Figure 8B) [130].

In vivo studies have shown poor brain penetration from both of these radiotracers. The synthesis of **[^11^C]22** was undertaken mainly in order to address the issue of the poor brain penetration of **[^11^C]18**, as it has lower lipophilicity (Log P = 3.3) than **[^11^C]18** (Log P = 3.8) and higher affinity binding (**[^11^C]22**
*K*i = 2.7 nM and **[^11^C]18**
*K*i = 3.3 nM). Nevertheless, the radioactivity in the brain of **[^11^C]22** was much lower than that of **[^11^C]18**, mostly due to its faster metabolization (high levels of **[^11^C]22** were present in the small intestine and liver). The poor brain penetration of **[^11^C]18** is mainly explained by the influence of the drug efflux transporters P-gp and BCPR. Therefore, in order to visualize the NPY Y_1_ receptors in the brain using this strategy, it is crucial to develop PET tracers with superior blood–brain barrier penetration and in vivo stability. Nevertheless, this study was useful in its provision of a better understanding of the in vivo properties of **18**.

Diverging from the classical methods of the radiolabeling of the Y_1_ antagonists, Vall-Sagarra et al. combined the use of heterobivalent peptidic ligands (HBPLs) with [^68^Ga] radiolabeling methods and developed a distinct approach to the development of new Y_1_ radiotracers [132]. As is known, radiolabeled peptides are a great tool for tumor visualizations as they target the many receptors that are overexpressed, though normally they target a specific receptor. As different tumor lesions can express different receptor types, the tumor visualization may not be as efficacious as it was previously hypothesized. Radiolabeled heterobivalent peptide ligands (HBPLs) have the capacity to specifically target more than one receptor type, rendering them advantageous when compared to monovalent peptides, enabling better tumor visualization and improved in vivo biodistribution. As was referenced earlier, human breast cancer overexpresses 85% of NPY Y_1_ receptor but, as a matter of fact, it also overexpresses the gastrin-releasing peptide receptor (GRPR) in about 75% of cases. Hence, the authors developed bispecific HBPLs and radiolabeled them with ^68^Ga and then proceeded to undertake further in vitro and in vivo characterizations in order to show the general feasibility of this distinct strategy. Thus, HBPLs **23**–**27** and monomeric reference peptides **28** and **29** (Figure 9) were synthesized and then radiolabeled with Ga^3+^. The ^68^Ga^3+^ was obtained via the elution of an ^68^Ge/^68^Ga generator system. Then, the pH of the solution was adjusted from 3.5 to 4.0 and the NODA-GA-comprising HBPLs **23**–**27** was incubated at 40–45 °C for 10 min. The DOTA-comprising reference compounds **28** and **29** reacted at 99 °C under identical conditions. The radiolabeled products **[^68^Ga]23**-**[^68^Ga]27**, **[^68^Ga]28** and **[^68^Ga]29** were attained in radiochemical yields and purities of 95–99% with molar activities of 10–15 GBq/µmol (used for the in vitro assays) or 40–46 GBq/µmol (used for the in vivo evaluations). The Log P of the different radiolabeled compounds was calculated and the results demonstrated the high hydrophilicity of all of the substances, a result which indicates that the in vivo pharmacokinetics of HBPLs **[^68^Ga]23**-**[^68^Ga]27** should be similar to those of the parent monomeric radiopeptides **[^68^Ga]28** and **[^68^Ga]29**. Moreover, regarding their stability, it was demonstrated that all of the compounds were stable over the testing period of 90 min. Cellular uptake studies were performed using the T-47D human breast cancer cell line as it was described to express both the GRPR and NPY Y_1_ receptors, allowing the observation of the binding of the synthesized HBPLs. The monovalent GRPR-specific peptide **[^68^Ga]28** displayed high and constant specific uptake in this cell line, as did HBPLs **[^68^Ga]23-[^68^Ga]27**. Nevertheless, the monovalent NPY Y_1_-binding peptide **[^68^Ga]29** showed no specific uptake, meaning that the uptake of the radiolabeled HBPLs was solely mediated by GRPR.

After the analysis of the in vitro studies’ results, the authors selected HBPL **[^68^Ga]25** to be used in the in vivo evaluations as, compared to the other HBPLs that were developed, it showed the highest stability and hydrophilicity as well as a slightly greater level of tumor cell uptake. Thus, small animal PET/CT imaging studies in T-47D tumor-bearing immunodeficient mice were performed using the HBPL **[^68^Ga]25** and also the scrambled variants: **[^68^Ga]25a** (PESIN_scrambled_ combined with [Lys^4^(aminooxy), Trp^5^, Nle^7^]BVD_15_) and **[^68^Ga]25b** (PESIN combined with [Lys^4^(aminooxy), Trp^5^ and Nle^7^]BVD_15,scrambled_), under identical conditions. The in vivo PET/CT imaging and ex vivo biodistribution data showed high kidney and liver uptakes. Regarding the tumor visualization, the scrambled analogs **[^68^Ga]25a** and **[^68^Ga]25b** were less efficient compared to the bispecific ligand **[^68^Ga]25**, a finding that indicates that both parts of HBPL **[^68^Ga]25** were necessary to enhanced in vivo tumor uptake and that this uptake was GRPR- and NPY Y_1_-specific. Concluding, the authors were able to demonstrate that the use of GRPR- and NPY Y_1_-specific HBPL were indeed beneficial to in vivo tumor uptake and visualization, compared to monospecific agents [132].

Recently, Krieger et al. have reported the synthesis of ^68^Ga-radiolabeled analogues of the most promising previously described truncated NPY analogues and compared them regarding their metabolic stability [133]. Of the nearly 80 truncated peptides analogues of NPY that have been developed over the years, the authors chose the following five by virtue of their metabolic stability: linear [Pro^30^,Lys(DOTA)^31^,Tyr^32^,Leu^34^]NPY_28–36_ ([Lys^4^(DOTA)]-BVD_15_), **30**; [Pro^30^,Lys(DOTA)^31^,Bip^32^,Leu^34^]NPY_28–36_, **31** (Bip: biphenylalanine); [Lys(lauroyl)^27^,Pro^30^,Lys(DOTA)^31^,Bip^32^,Leu^34^]NPY_27–36_, **32**; Ac[D-Cys^29^,Lys(DOTA)^31^,Cys^34^]NPY_29-36_, **33** and heterodimer [Pro^30^,Cys^31^,Trp^32^,Nle^34^]NPY_28-36_-[Lys(DOTA)^29^,Pro^30^,Cys^31^,Trp^32^,Nle^34^]NPY_28–36_, **34**. The monomers **30**–**33** and heterodimer **34** were radiolabeled with ^68^Ga^3+^ (which was obtained by the fractioned elution of an IGG ^68^Ge/^68^Ga generator system) in an aqueous acetate-buffered solution. The radiolabeling reaction occurred at 99 °C, for a duration of 10 min at a pH level of 3.5–4.0. The products **[^68^Ga]30**-**[^68^Ga]32** and **[^68^Ga]34** were obtained in high radiochemical yields and purities of 96–99% and with non-optimized molar activities of 18.8–23.1 GBq/μmol, starting from 376.6–461.0 MBq of ^68^Ga^3+^. Pure **[^68^Ga]33** could not be obtained through these reaction conditions due to the formation of a significant number of side products. Despite the authors’ efforts to find out the origin of those side products, the origins remained inconclusive and so further **[^68^Ga]33** characterization and stability evaluation was omitted. The Log P values were determined and they showed an increasing variation from the hydrophilic **[^68^Ga]30**, with a Log P value of −3.61 ± 0.37, to lipophilic **[^68^Ga]32**, with a Log P value of −0.48 ± 0.14 and the heterodimer **[^68^Ga]34** showed a hydrophilic character as well, with a Log P value of −2.47 ± 0.18. The stability of these four radioligands was determined by the use of a human serum and human microsomal stability assay. Regarding the human serum stability assay, the radioligands showed the formation of different hydrophilic metabolites and considerable differences concerning their rate of degradation by peptidases, which caused a great difference in the half-lives of the compounds. **[^68^Ga]30** showed a low stability in the serum with a half-life of 20 min. The modification of the Tyr^32^ amino acid by artificial Bip in **[^68^Ga]31** led to a higher half-life of this radioligand, which amounted to 65 min, ensuring higher stabilization against degradation. The additional N-terminal lauroylation in **[^68^Ga]32** further increased the serum stability of the radiotracer, resulting in a half-life of 144 min, one which is more than adequate for ^68^Ga-PET imaging uses as PET scans using ^68^Ga are usually performed within the first 60 to 90 min post injection. Comparable to **[^68^Ga]31**, **[^68^Ga]34** showed a half-life of 67 min. Regarding the liver microsomal stability assay, the same stability trend was found within the radiopeptides **[^68^Ga]30**-**[^68^Ga]32**, with **[^68^Ga]32** being the most stable among them. Heterodimer **[^68^Ga]34** was degraded more rapidly by the liver enzymes than in the human serum assay, presenting a half-life of only 6 min. Thus, [^68^Ga][Lys(lauroyl)^27^,Pro^30^,Lys(DOTA)^31^,Bip^32^,Leu^34^]NPY_27-36_ (**[^68^Ga]32**) was the most promising truncated NPY analogue for the development of peptide-based NPY Y_1_ receptor imaging agents within this study. It showed excellent in vitro receptor binding affinity and a very high level of stability towards proteolytic degradation by peptidases, both in human serum and in the human liver.

More recently, still focusing on in vivo receptor targeting with radiolabeled peptide-based probes, Cardoso et al. described the development and characterization of two ^68^Ga-labeled NPY analogues for their potential use in breast cancer diagnosis [134]. Both of the analogues shared the same amino acid sequence: Tyr-Arg-Leu-Arg-BPA-Nle-Pro-Asn-Ile (BPA: L-4-benzoylphenylalanine), one which favors preferential binding to NPY Y_1_ receptors, and were derivatized with NOTA (Figure 10) through a lysine (**35a**) or an acetylated lysine (**35b**) linker.

The authors assessed different labelling reaction conditions, such as various peptide masses (50, 100 or 150 µg), incubation temperatures (20 and 95 °C) and reaction times (5 and 10 min). The best radiolabeling conditions were found to be 100 µg of ligand, a pH of 4.5, 5 min incubation time and 95 °C incubation temperature. With these conditions, both radiopharmaceuticals **[^68^Ga]35a** and **[^68^Ga]35b** showed radiochemical purities that were higher than 95% and neither gallium colloids nor ^68^GaCl_3_ were detected. Afterwards, the physicochemical properties were evaluated. Both of the complexes were hydrophilic, with Log P values of −3.2 ± 0.1 for **[^68^Ga]35a** and −2.6 ± 0.1 for **[^68^Ga]35b**, though the authors argue that these Log P values should not negatively affect the cellular uptake of the complexes as it depends on the expression of the NPY receptors on the cellular surface. In general, both of the complexes showed great in vitro stability, good cellular uptake, binding affinities in the nanomolar range and high cellular internalization rate. Interestingly, both of the complexes showed significant differences in their cellular uptake and externalization rates, as well as their Log P values. As the complex **[^68^Ga]35b** has a modified lysine, this makes it less hydrophilic and, consequently, increases its cellular uptake and reduces its externalization rate. Regarding their biodistribution profiles, both of the radiolabeled compounds had a high rate of blood depuration and renal excretion, being consistent with their hydrophilicity. However, the tumor uptake was moderate and ratio of tumor-to-muscle and tumor-to-blood were not as high as those which have been reported in the literature as being the most promising for diagnostic imaging. Nonetheless, these studies have shed light on the influence of the labelling technique in the development of new promising radiopharmaceuticals.

### 4.2. NPY Y_2_ Receptor

Winterdahl et al. reported the development of the first and, to date, the only PET radiotracer for the Y_2_ receptor [135]. The authors took advantage of the suitable characteristics of the Y_2_ receptor antagonist JNJ-31020028 (**36**, Figure 11) and radiolabeled this compound using [^11^C]CH_3_I in order to visualize the Y_2_ receptors in the living brain. JNJ-31020028 (*N*-(4-{4-[2-(diethylamino)-2-oxo-1-phenylethyl]piperazin-1-yl}-3-fluorophenyl)-2-pyridin-3-ylbenzamide) is a Y_2_ receptor antagonist with high affinity and selectivity for the receptors, poor oral availability (6%) in rats but high bioavailability subcutaneously (100%) and a half-life of 0.83 h. It penetrates the blood–brain barrier and occupies 90% (at 10 mg/kg) of the Y_2_ receptors [76]. The radiosynthesis of *N*-[^11^C]-methyl-JNJ-31020028 (**[^11^C]37**) was performed in an semiautomated system wherein [^11^C]methyl iodide was produced from [^11^C]CO_2_. [^11^C]CH_3_I was trapped in a solution of **36** and NaH dissolved in DMF and THF. The reaction occurred over the course of 3 min at 50 °C (Figure 11). The crude product was then purified by preparative HPLC and the radiolabeled compound **[^11^C]37** was obtained with radiochemical purity > 98% and molar activity > 100 GBq/µmol (the radiochemical yield was not disclosed).

The novel PET tracer **[^11^C]37** was then assessed by the use of autoradiography in brain sections and by the in vivo PET imaging of the pig brain. The radiographic results have demonstrated that high densities of **[^11^C]37** were found in the hippocampus and cerebellum, this is a finding which agrees with the anatomical distribution of the Y_2_ receptors that has been described in other animal species. The radiotracer **[^11^C]37** was also shown to be distributed rapidly into the brain and metabolized slowly in the bloodstream. The authors also concluded that PET studies using cyclosporine improved the target-to-background ratio of **[^11^C]37**, enabling the estimation of the pharmacokinetic parameters. This is due to the inhibition of the efflux transporter P-gp by cyclosporine as PET studies without cyclosporine gave a value of less than 1 for the ratio between the area under the whole-brain time–activity curve and the plasma time–activity curve of the radiotracer **[^11^C]37**, suggesting efflux transport at the blood–brain barrier [135].

Furthermore, Andersen et al. described the [^11^C]carbonyl labeling of the same compound, JNJ-31020028 (**36**), by using palladium-aryl oxidative addition complexes in carbonylation reactions with [^11^C]carbon monoxide [136]. [^11^C-Carbonyl]JNJ-31020028 (**[^11^C]36**) was achieved with the best radiochemical yield of 25% (Table 5) and radiochemical purity of 55%, when the reaction of the amino precursor **38** was undertaken with the XantPhos ligated aryl palladium complex **39** and guanidine base Me-TDB (6 equiv.) in dioxane, at 95 °C for 10 min. To this end, the uses of different bases and ligated aryl palladium complexes and their correspondents ^11^CO trapping efficiency and **[^11^C]36** radiochemical purity and yield were evaluated. In a first attempt, when NEt_3_ (1 equiv.) was added to the reaction of the amino precursor **38** with PPh_3_ ligated aryl palladium complex, no traces of ^11^CO trapping were observed (Table 5, entry 1). The implementation of the XantPhos ligated aryl palladium complex allowed the formation of the desired product **[^11^C]36**, though with low radiochemical yield, when DBU (4 equiv.) was used in the reaction (Table 5, entry 2). An increase in the reaction temperature led to a higher ^11^CO trapping efficiency but lower radiochemical purity of the product (Table 5, entry 3). A series of different organobases were also tested. The use of DBN, proton-sponge and TMGN bases (4 equiv.) led to a low level of ^11^CO gas trapping (Table 5, entries 4–6); however, the use of the guanidine base Me-TDB (4 equiv.) increased the reactivity profoundly and significantly higher values for the trapping efficiency and radiochemical purity of the product were observed (Table 5, entry 7). The augmentation of the temperature of the reaction did not have a beneficial impact (Table 5, entry 8). Finally, the increase in the guanidine base amount to 6 equivalents led to the formation of [^11^C-carbonyl]JNJ-31020028 (**[^11^C]36**) with a good purity profile and acceptable radiochemical yield on an average of three reproducible reactions (Table 5, entry 9).

### 4.3. NPY Y_5_ Receptor

In the literature, the first reported research using a PET ligand for the Y_5_ receptor, performed by Erondu et al. [42], aimed to test if NPY Y_5_ antagonism would lead to weight loss in overweight and obese patients. In order to verify this hypothesis, MK-0557 [42] (**40**, Figure 12), a potent selective and orally available Y_5_ antagonist, was used. In this case, these PET studies were aimed at providing complementary data about the MK-0557 dosage by studying the Y_5_ receptor occupancy after the oral administration of **40**. The selective Y_5_ PET ligand [^11^C]MK-0233 (**[^11^C]41**), which had been developed and validated in the rhesus monkey [137], was used in human volunteers. **[^11^C]41** has a similar structure to the Y_5_ antagonist **40** and it is also extremely selective to this receptor (Figure 12). The long-term weight loss study with MK-0557 showed that, even though this NPY Y_5_ antagonist had a favorable clinical safety profile, the magnitude of the induced weight loss was not clinically meaningful as the degree of weight loss after 52 weeks of treatment with **40** was significantly less compared to other weight loss drugs, the report showed [42]. These clinical studies provided a better clarification of the role of the NPY Y_5_ receptor in human energy homeostasis and its utility as a target for anti-obesity drug therapy. The authors also concluded that, for the purposes of future anti-obesity drug development programs, targeting only the NPY Y_5_ receptor was unlikely to produce therapeutic efficacy [42].

The synthesis and radiolabeling of compound **41** with [^11^C] was later published by Takahashi et al. [138]. The authors reported a series of *trans*-3-oxospiro[(aza)isobenzofuran-1(3*H*),1′-cyclohexane]-4′-carboxamide derivatives and examined their binding affinities towards the Y_5_ receptor and their brain penetrability. Compound **41** showed the most promising biological profile for becoming a Y_5_ PET tracer as it had a good Y_5_ binding affinity of 1.5 ± 0.3, an acceptable Log P value of 2.79 and was selective to the Y_5_ receptor (human Y_1_, Y_2_ and Y_4_ IC_50_ > 10 µm). Moreover, its susceptibility to human and mouse P-glycoprotein transporters was studied and it was concluded that the precursor **41** was not a substrate of the human P-gp transporter. Thus, compound **41** was selected to be radiolabeled with [^11^C] in order to become a novel Y_5_ PET tracer. Succinctly, the synthesis of **41** starts with the reaction of 3-bromopyridine **42** with LDA followed by 1,4-cyclohexanedione mono-ethyleneketal and acid mediated hydrolysis so as to obtain the ketone **43** (Figure 13). This was then stereoselectively reduced by NaBH_4_, providing the *cis*-alcohol **44**. The *trans* carboxylic acid **45** was achieved by the mesylation of alcohol **44**, followed by the reaction with NEt_4_CN and acid mediated hydrolysis. Compound **45** was then coupled with 2-amino-5-*o*-fluorophenylpyrimidine in order to afford the precursor **46**. The subsequent radiolabeling of precursor **46** can then be achieved by its reaction with [^11^C]CO in the presence of Pd(PPh_3_)_4_, providing **[^11^C]41** as a product [137,138]. No statements of the yield or molar activity of the radiolabeled product were disclosed.

Afterwards, Kealey et al. reported the [^11^C]carbonylation reaction of **[^11^C]41** using microfluidic technology [139]. The authors sought to enhance the reactivity of [^11^C]CO by improving its solubility via chemical complexation to CO-binding molecules in a solution. They were able to achieve this by forming a copper [^11^C]carbonyl complex, copper(I)tris(3,5-dimethylpyrazolyl)borate-[^11^C]carbonyl (Cu(Tp*)[^11^C]CO), which was then used in the [^11^C]carbonylation reaction that led to the formation of the radiolabeled compound **[^11^C]41**. Hence, the radiosynthesis of **[^11^C]41** was produced through the ring-closing [^11^C]carbonylation reaction between pyridyl-bromide and the hydroxyl functional groups of the precursor **46** and mediated by the catalyst Pd(PPh_3_)_4_. In this work, the authors synthesized **[^11^C]41** in two different ways: one using a microfluidic methodology and one using a low-pressure technique. Regarding the first method, the authors analyzed the temperature and flow rate’s influence on the radiolabeling reaction and, so as to maximize the yields and minimize the synthesis reaction time that were required in order to obtain enough radiotracer **[^11^C]41** to perform the PET studies, the reactions were performed at 180 °C with a flow rate of 100 µL/min. This led to the production of [^11^C]MK-0233 with a molar activity of 100 ± 30 GBq/µmol, radiochemical purity > 99% and a decay-corrected radiochemical yield of 7.2 ± 0.7% in 27 min of synthesis time. Concerning the second method, a 3 mL vial was used as the reaction vessel and, despite some problems and subsequent optimizations related to the apparatus, the radiolabeled compound was achieved with a molar activity of 100 ± 15 GBq/µmol, radiochemical purity > 99% and a decay-corrected radiochemical yield of 7.1 ± 2.2%, also in 27 min. The authors had initially predicted a greater difference between the results of these methods, but the radiochemical yields were almost identical and they were produced in the same amount of time. Nevertheless, the feasibility of using microfluidics for solutions-based carbonylations proved it to the be a suitable method for PET tracers’ production, with the potential to surpass the vial method.

More recently, Kumar et al. reported the identification and radiolabeling of four potent and selective Y_5_ receptor antagonists that could emerge as Y_5_ receptor PET tracers. The selected four candidates (**47**, **48**, **49** and **50**, Figure 14) showed high affinities towards the receptor and were suitable for radiolabeling with [^18^F]fluoride via nucleophilic substitution reactions [140].

The authors performed the non-radiolabeling experiments using KF/krytofix/K_2_CO_3_ in DMSO at 180 °C. Regarding the radiolabeling experiments, both conventional heating and microwave irradiation were tested. The radiolabeling of the compound **47** did not provide any radiolabeled product under conventional heating, although under microwave irradiation **[^18^F]47** was obtained in <2% isolated RCY (Table 6). The same results were obtained when radiolabeling compound **49**. [^18^F]LuAE00654 (**[^18^F]48**) was obtained with isolated RCY of 10 ± 4% and the radiolabeling of compound **50** produced **[^18^F]50** with 25 ± 5% RCY under microwave irradiation and 15 ± 3% RCY under conventional heating conditions. The radiochemical purities of the obtained radiolabeled products were >95% with an average molar activity of 2.5 ± 1% Ci/µmol. Given these results, PET studies were performed with the 2-substituted pyridyl thiazoles [^18^F]LuAE00654 (**[^18^F]48**) and **[^18^F]50** as they achieved better radiochemical yields than 6-substituted pyridyl thiazoles. The PET studies were carried out in fasted adult male baboons (*Papio anubis*). **[^18^F]48** showed good BBB penetration and was retained in several brain regions such as the caudate, putamen and cortical regions. The distribution of this radiotracer was shown to be compliant with the acknowledged distribution of the Y_5_ receptor and [^11^C]MK0233 in monkeys and human subjects. [^18^F]LuAE00654 also demonstrated itself to be of rapid clearance and its specificity was proven by blocking experiments using the Y_5_ ligand LuA44608. Concerning the radiotracer **[1^8^F]50**, even though it penetrated the baboons’ blood–brain barriers, it showed little retention of radioactivity in the brain, fast washout and a faster metabolization in comparison to [^18^F]LuAE00654 (**[^18^F]48**). Thus, [^18^F]LuAE00654 (**[^18^F]48**) showed great promise as a PET radiotracer for the Y_5_ receptor and could be used for the development of novel drugs aiming at the NPY Y_5_ receptor functions.

## 5. Conclusions

NPY has gained a lot of interest from the scientific community over the years and has been broadly studied and biologically and pharmacologically characterized due to its involvement in the pathophysiology of several diseases, especially brain diseases. To further understand its functioning, PET imaging of the NPY receptors has been investigated. For that purpose, several radiotracers were developed for three of the NPY receptors (Y_1_, Y_2_ and Y_5_). In this paper, we have highlighted some approaches to the chemical and radiolabeling development of the different radiotracers for the NPY receptors and the subsequent PET studies and results, which were mainly used in tumor visualization and receptor visualization associated with obesity and other hypothalamic disorders. 

Despite the promising results that have been obtained so far with the pre-clinical and clinical studies, which were crucial to the better understanding of the NPY receptors in the several pathologies, it is clear that further investigation is still needed. The advances in NPY system PET imaging have been occurring at a slower rate as there is still some difficulty in not only developing novel radiotracers that combine good radiolabeling methods with good pharmacological profiles but also in expanding the preclinical development to the clinical area. Nevertheless, PET imaging of the NPY system holds a lot of potential to further the developments in and insights into this topic and to support the discovery and development of new promising CNS drugs that are able to target this important family of receptors in the years to come.

## Figures and Tables

**Figure 1 molecules-27-03726-f001:**
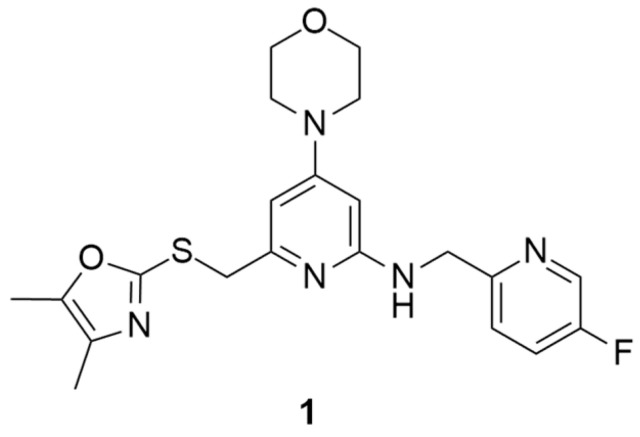
Structure of Y_1_ receptor antagonist **1**.

**Figure 2 molecules-27-03726-f002:**
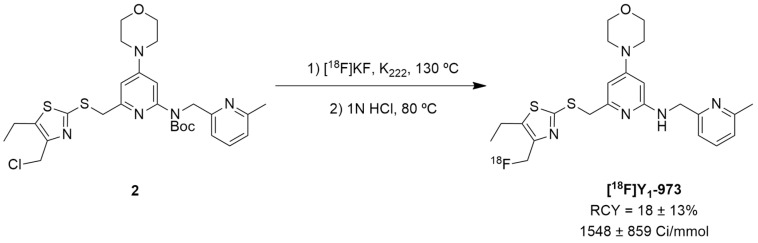
The radiosynthesis of the NPY Y_1_ PET tracer [^18^F]Y_1_-973.

**Figure 3 molecules-27-03726-f003:**
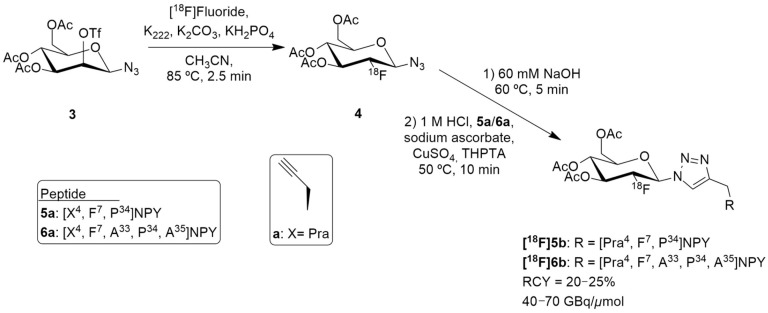
Radiosynthesis of **[^18^F]5b** and **[^18^F]6b**.

**Figure 4 molecules-27-03726-f004:**
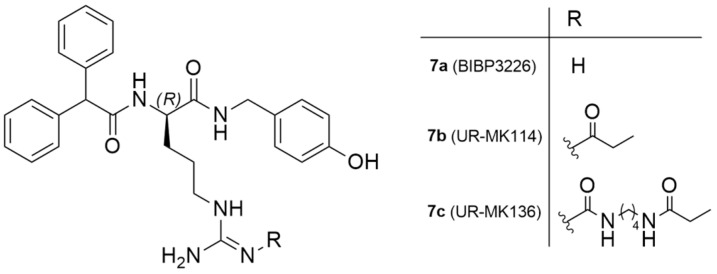
Chemical structure of Y_1_ receptor antagonist BIBP3226 and Y_1_ ligands derived from argininamide-type antagonists.

**Figure 5 molecules-27-03726-f005:**
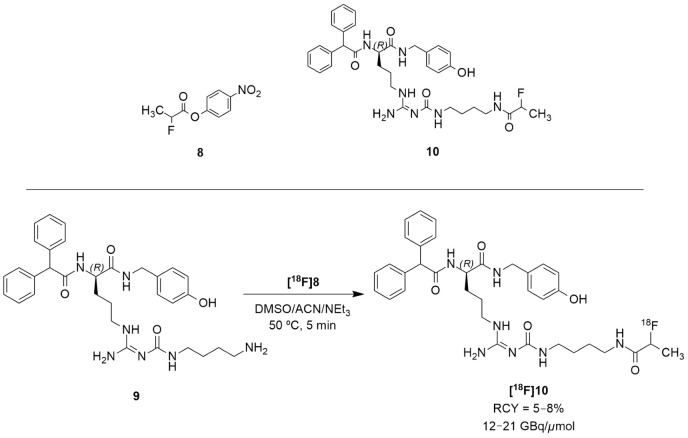
Radiosynthesis of **[^18^F]10**.

**Figure 6 molecules-27-03726-f006:**
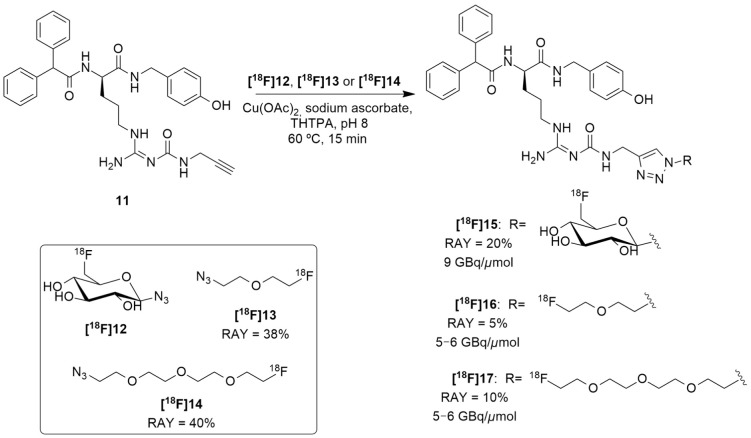
Radiosynthesis of **[^18^F]15**, **[^18^F]16** and **[^18^F]17**.

**Figure 7 molecules-27-03726-f007:**
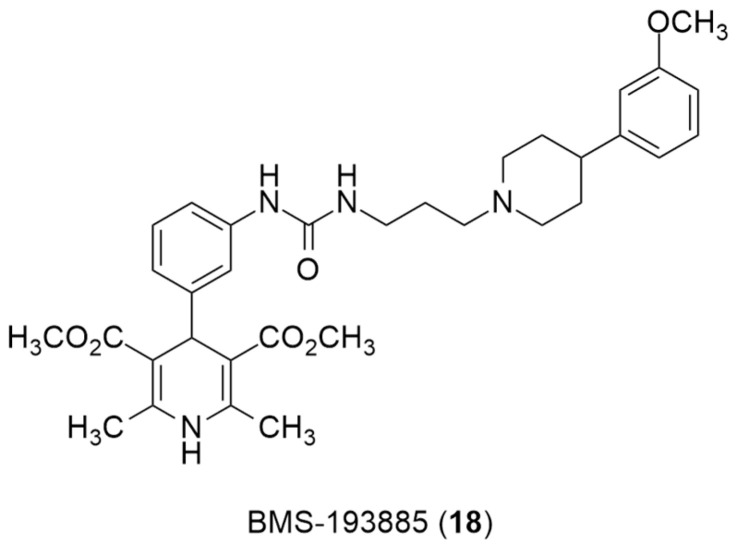
Chemical structure of Y_1_ receptor antagonist BMS-193885(**18**).

**Figure 8 molecules-27-03726-f008:**
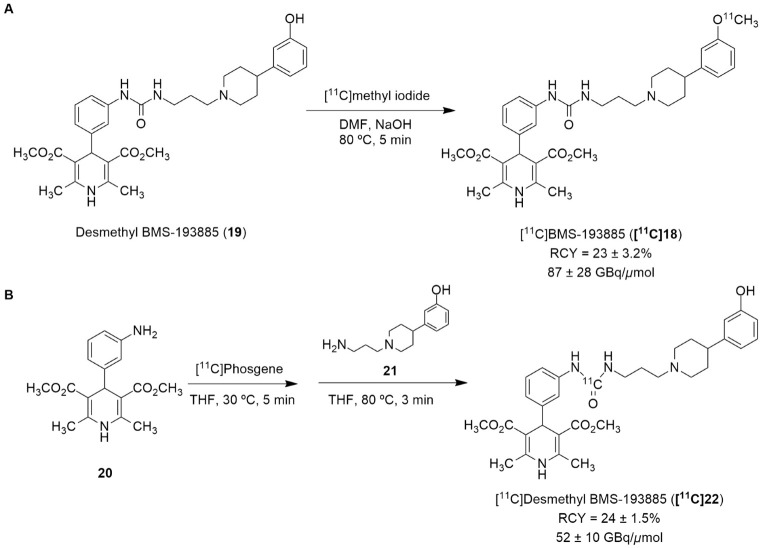
Radiosynthesis of **[^11^C]19** via [^11^C]CH_3_I (**A**) and **[^11^C]22** via [^11^C]phosgene (**B**).

**Figure 9 molecules-27-03726-f009:**
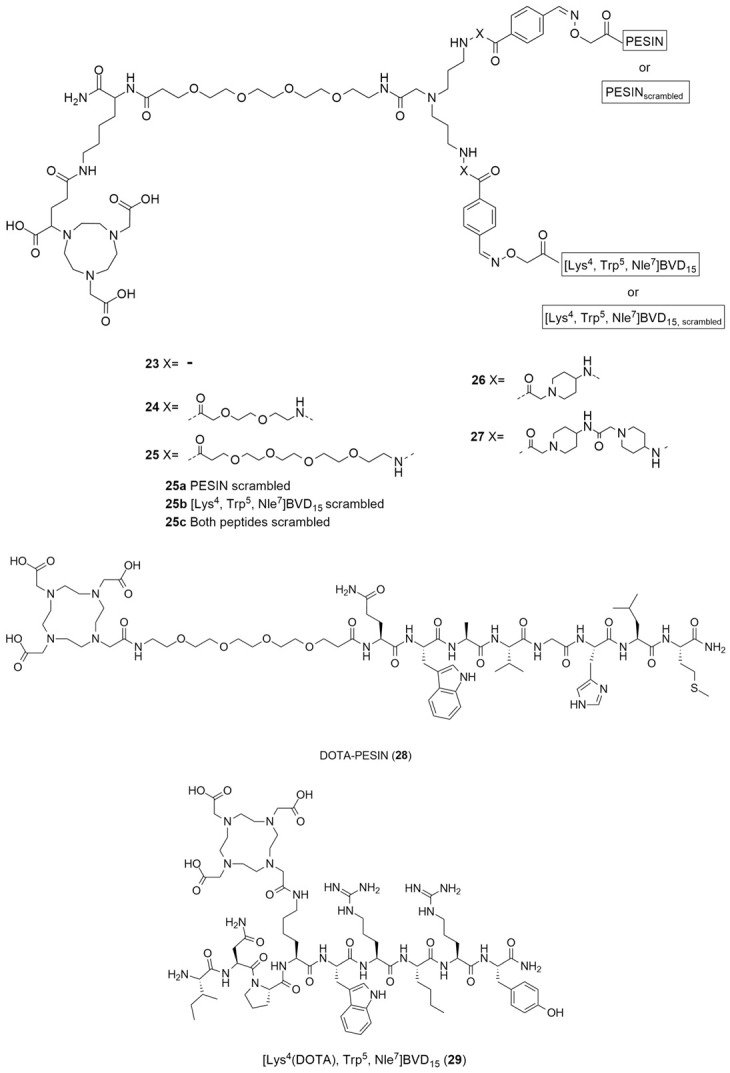
Structures of HBPLs **23**–**27**, DOTA-PESIN (**28**) and [Lys^4^(DOTA), Trp^5^, Nle^7^]BVD_15_ (**29**) as mono-specific reference substances for the HBPLs in in vitro tumor cell uptake studies.

**Figure 10 molecules-27-03726-f010:**
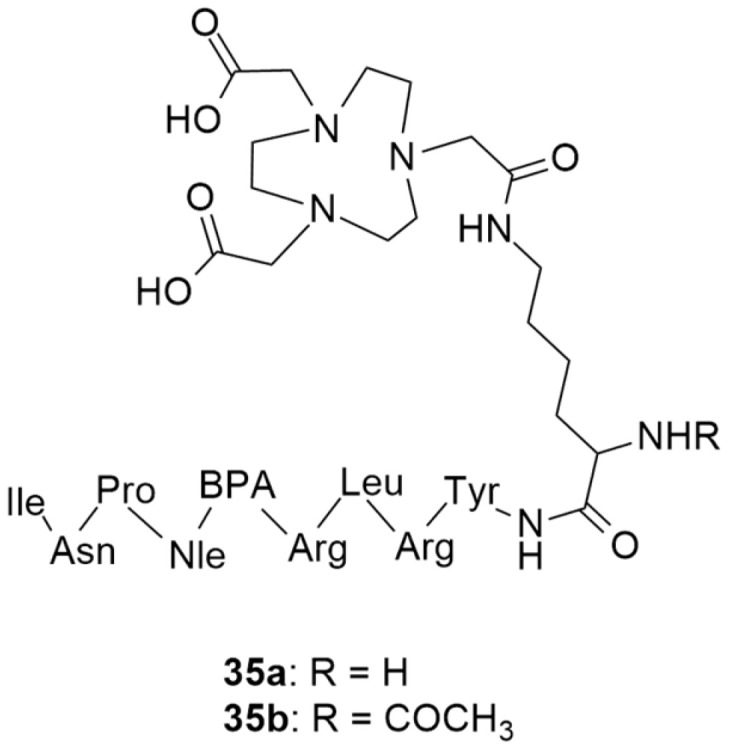
Structures of short NPY analogues functionalized with NOTA (**35a** and **35b**).

**Figure 11 molecules-27-03726-f011:**
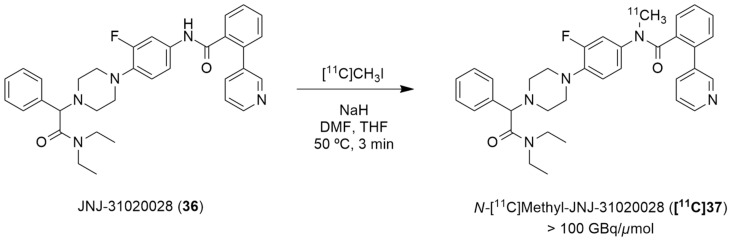
Radiosynthesis of N-[^11^C]methyl-JNJ-31020028 (**[^11^C]37**).

**Figure 12 molecules-27-03726-f012:**
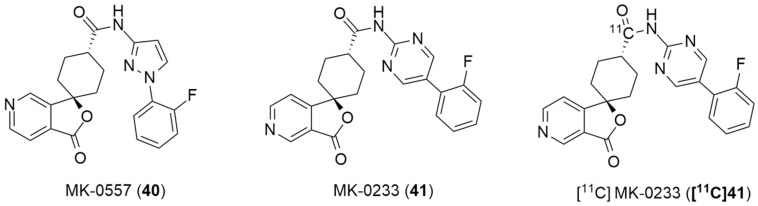
Structures of MK-0557 (**40**), MK-0233(**41**) and PET ligand [^11^C]MK-0233 (**[^11^C]41**).

**Figure 13 molecules-27-03726-f013:**
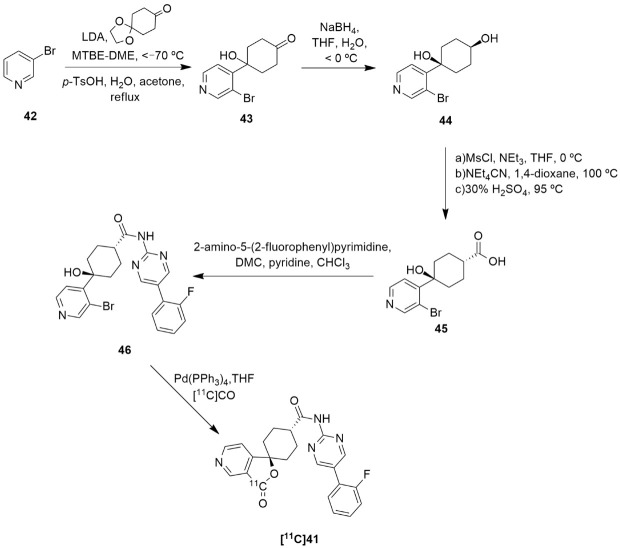
Synthetic route of precursor **46** and radiosynthesis of **[^11^C]41**.

**Figure 14 molecules-27-03726-f014:**
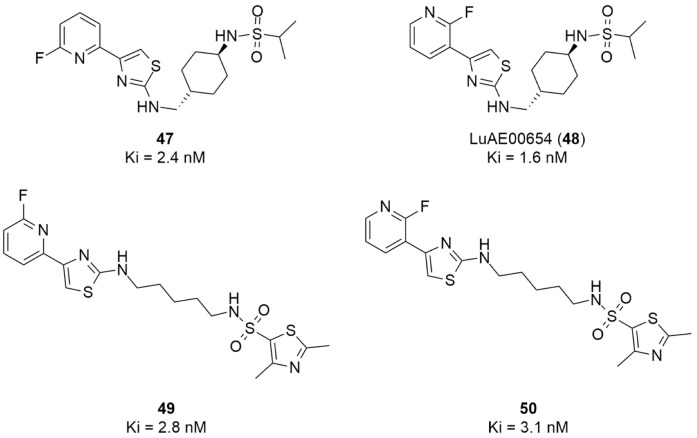
Selected candidate PET ligands for the Y_5_ receptor.

**Table 1 molecules-27-03726-t001:** Properties and characteristics of the following imaging modalities: CT, MRI, SPECT and PET.

Modality	Characteristics	Advantages	Limitations
CT	▪ X-rays ▪ Spatial resolution: 0.5 mm	▪ Scanning is fast, painless and non-invasive ▪ High spatial resolution ▪ Ability to image bone, soft tissue, and blood vessels all at the same time	▪ Ionizing radiation ▪ Requires contrast agent ▪ Radiation tissue nonspecificity
MRI	▪ Spatial resolution: 1 mm	▪ Non-invasive ▪ High spatial resolution and soft tissue contrast ▪ Lack of ionizing radiation	▪ Low sensitivity ▪ Lack of chemical specificity ▪ High cost
SPECT	▪ Radionuclides (^123^I, ^99m^Tc, ^111^In, ^67^Ga) ▪ Spatial resolution: 6 mm	▪ SPECT images can provide information on physiological and physiopathological processes at a molecular level ▪ Non-invasive	▪ Its images are less sensitive and less detailed than PET images ▪ Less expensive than PET
PET	▪ Radionuclides (^11^C, ^13^N, ^18^F, ^68^Ga, ^64^Cu) ▪ Spatial resolution: 4 mm	🞍 PET images provide physiological and detailed metabolic information ▪ Better spatial resolution and higher sensitivity than SPECT ▪ Non-invasive	▪ Cyclotron needed ▪ High cost

**Table 3 molecules-27-03726-t003:** SAR of compounds **1a**–**e**.

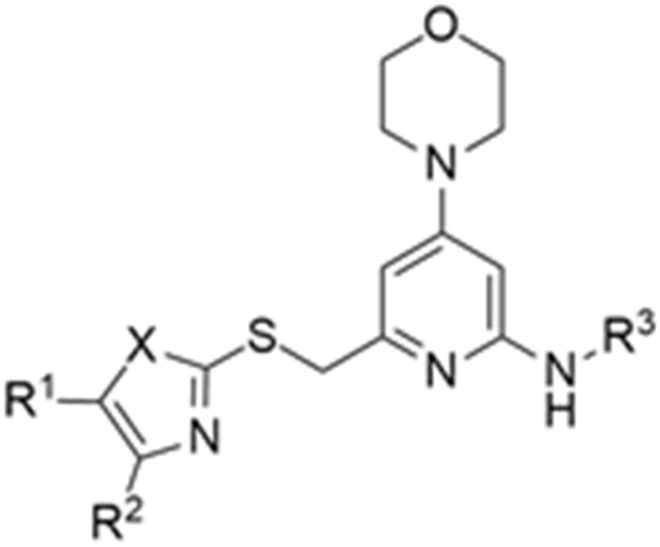						
Compound	X	R^1^	R^2^	R^3^	Y1 Binding Affinity IC_50_ (nM)	Log P
**1a**	O	CH_3_	CH_3_	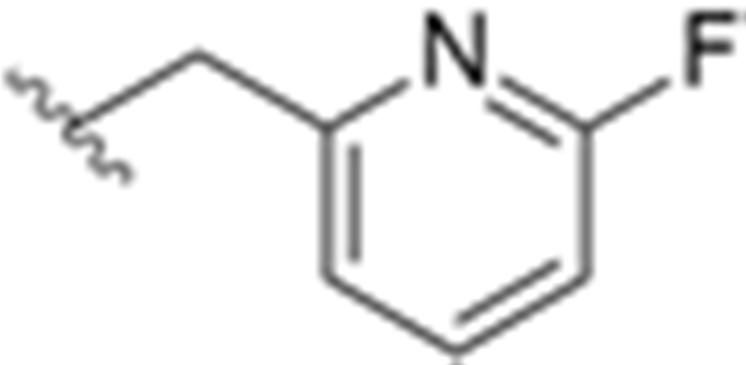	1.5	3.1
**1b**	O	CH_3_	CH_3_	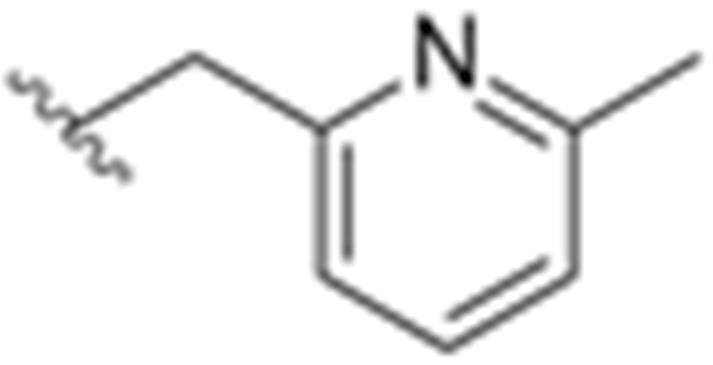	0.69	2.8
**1c**	S	CH_2_F	CH_3_	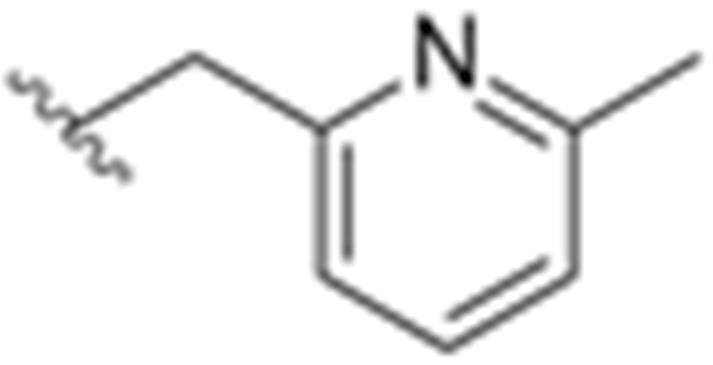	0.56	2.3
**1d**	S	CH_3_	CH_2_F	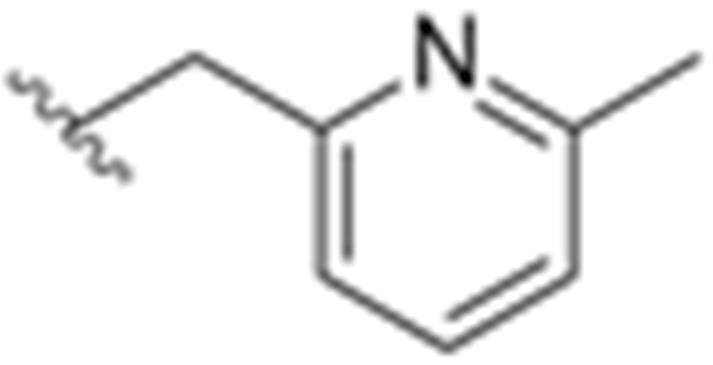	0.20	2.7
**1e**	S	CH_2_CH_3_	CH_2_F	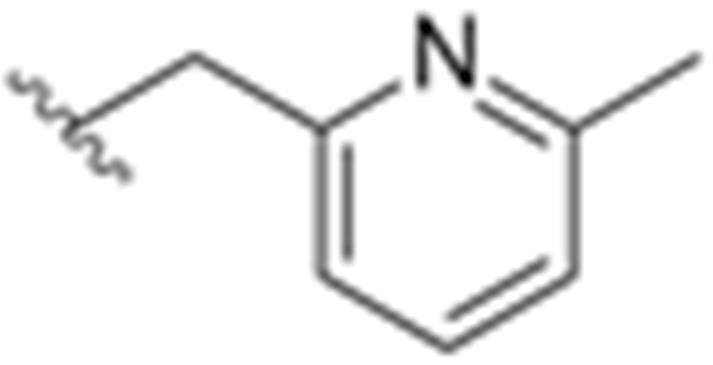	0.13	3.2

**Table 4 molecules-27-03726-t004:** Structure and in vitro properties of compound **1d** and **1e** (Y1-973).

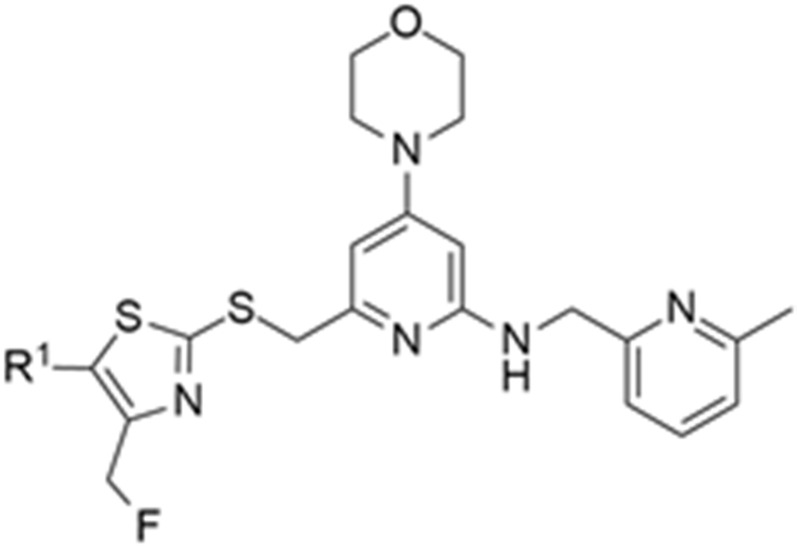					
Compound	R^1^	Y_1_ IC_50_ (nM)	Y_2_, Y_4_, Y_5_ IC_50_ (µM)	Log P	P-gp Transport Ratio
**1d** [123]	CH_3_	0.20	>10	2.7	1.8
**1e** (Y1-973) [123,124]	CH_2_CH_3_	0.13	>10	3.2	1.4

**Table 5 molecules-27-03726-t005:** Compound JNJ-31020028 [^11^C]-Carbonyl labeling.

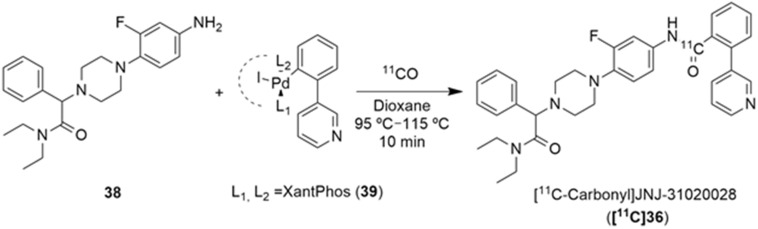					
Entry	Complex	Base	Trapping Efficiency (%)	Radiochemical Purity (%)	Radiochemical Yield (%)
1	L_1_ = L_2_ = PPh_3_	TEA	-	-	-
2 ^a^	**39**	DBU	16	23	4
3 ^b^	**39**	DBU	36	10	4
4	**39**	DBN	15	3	0.5
5	**39**	Proton-sponge	6	-	-
6	**39**	TMGN	22	-	-
7	**39**	Me-TDB	38	55	21
8 ^c^	**39**	Me-TDB	42	25	11
9	**39**	Me-TDB	48 ± 7	55 ± 7	25 ± 4 (*n* = 3)

^a^ 90 °C; ^b^ 100 °C; ^c^ 115 °C

**Table 6 molecules-27-03726-t006:** Radiosynthesis of NPY Y5 ligands under microwave irradiation.

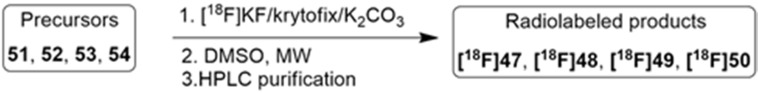
Precursor	Radiolabeled Product	RCY
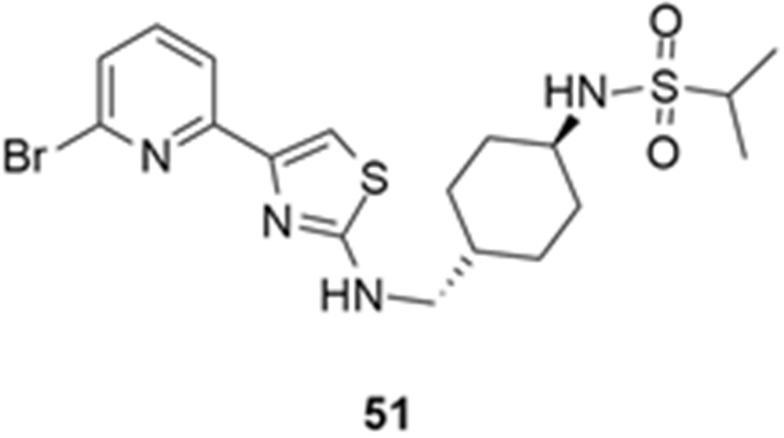	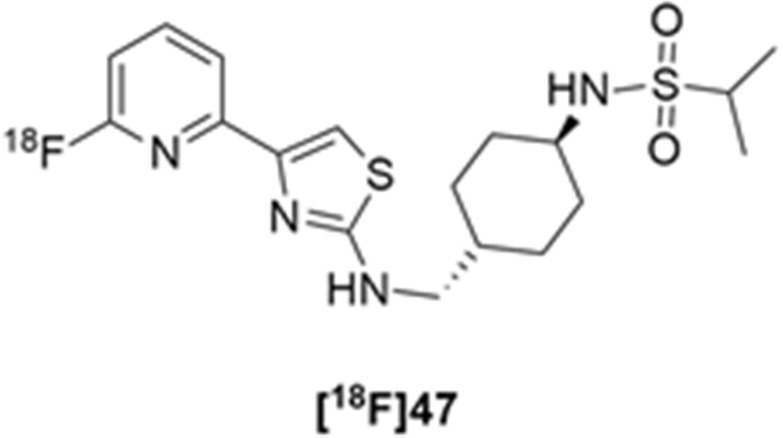	<2%
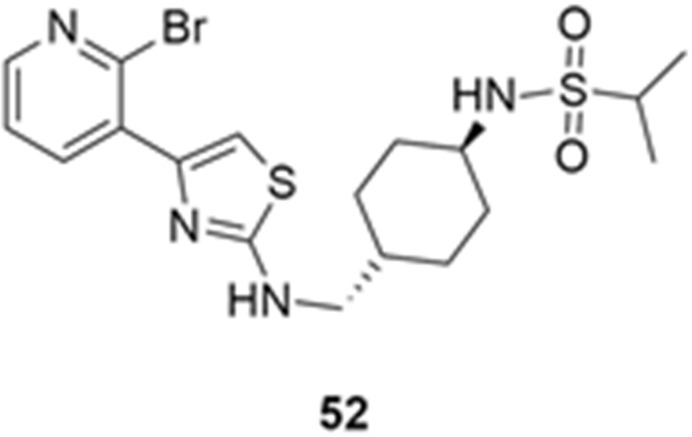	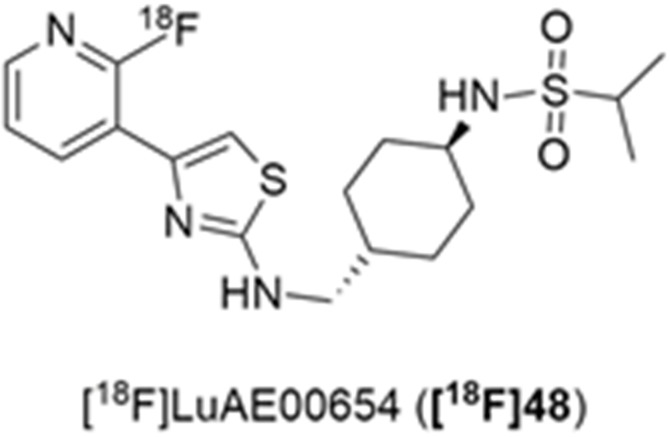	10%
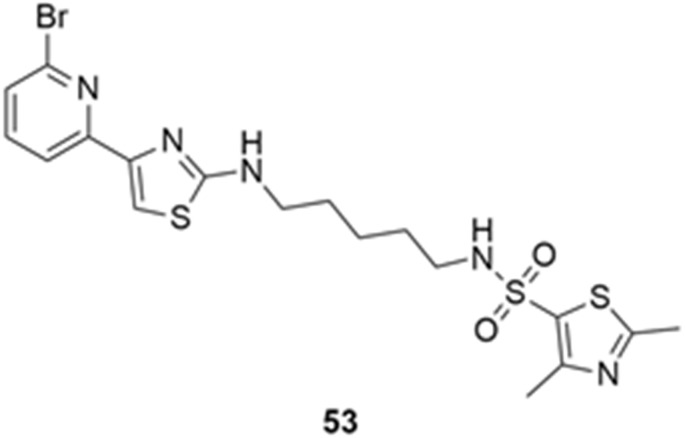	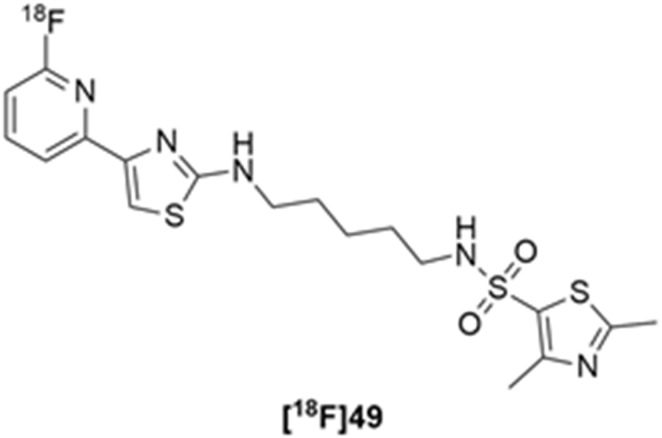	<2%
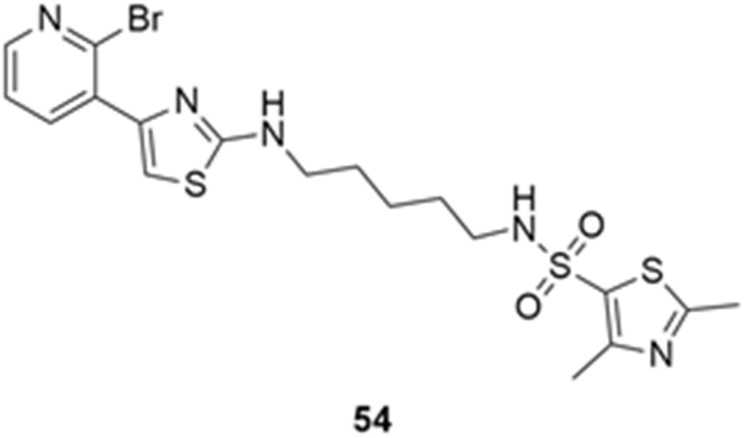	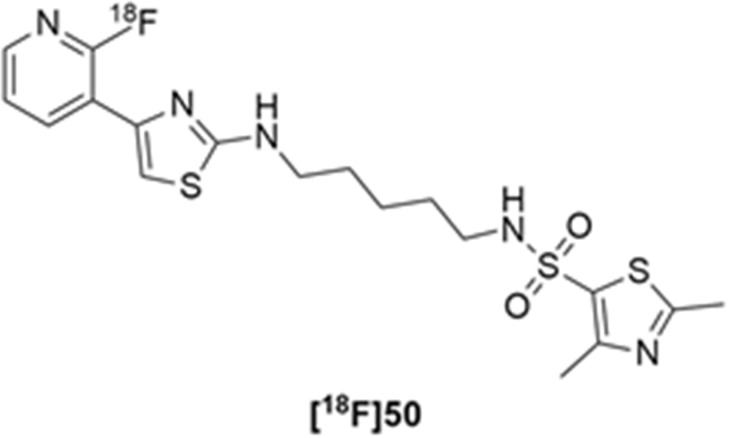	25%

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
