# Peer review of "PET Imaging of the Neuropeptide Y System: A Systematic Review"

_molecules, 2022, doi:10.3390/molecules27123726_

Round 1

Reviewer 1 Report

In this review article, Fonseca and colleagues review the development of radiotracers to image the neuropeptide y system. The review is thorough and informative and deserves to be published after addressing some minor points.

P2 L65- “NPY is strictly localized to neurons” – isn’t NYP released by neurons and therefore no longer localized to neurons?

Fig 3. X = Pra – use full name or define it. The structure shown in Fig 3 inset is confusing as it has unclear the connectivity

Fig 13 is missing reactants

- The article discusses in multiple occasions the brain penetration of the different tracers as well as whether they are substrates of the P-gp efflux pump but for the most part it does not discuss how they get into the brain. Describing which agents they get into the brain by passive diffusion and which ones get by active transport would be useful for the readers.

The final conclusion is vague: “further studies are needed”. Please be more specific as to what is needed (eg. tracers with higher affinity, higher brain penetration, higher selectivity, better labeling methods??)

Author Response

We thank the reviewer for the helpful comments. All changes were implemented, with the exception of the request of discussion of the mechanism for brain penetration. Even though an effort was made to find this information in publish literature, we did not find. Authors use the PET images as a proof of brain penetration, without further discussion.

Reviewer 2 Report

The paper on the role of PET in the imaging of neuropeptide Y presents a topical subject and the authors managed to write a fairly comprehensive review though based on relatively old articles. I have a number of comments that the authors should address to improve the quality of the manuscript:

(1)   The authors state that this is a systematic review of the literature, though it reads more like a narrative review. What databases were used for the article selection? For a systematic review, the authors must supply a PRISMA diagram to illustrate the selection criteria for the scientific articles discussed in this work.

(2)   Also, a section on Methodology – search criteria must be included in the manuscript, detailing the number of articles found on the topic and the articles that are presented in the review.

(3)   While PET has indeed an important role in neuropeptide Y system imaging, there are other techniques (such as SPECT/CT) which might be more cost effective but also present some drawbacks. This aspect (i.e., of other imaging techniques) should be discussed in the review, in a comparative manner with PET.

(4)   Most articles referenced in this paper are old. Of the 137 articles itemized in the reference list, only 5 (<4%!) are published in the last 5 years. This is not to say that the authors have dismissed the latest research, as the number of recent publications is small indeed. Nevertheless, the authors should make a comment on this aspect as well – why is the research on neuropeptide imaging advances so slowly?

A recent reference (clinical trial) that the authors have missed:

https://pubmed.ncbi.nlm.nih.gov/34305816/ 

Specific comments:

Line 43 – replace ‘fentomolar’ with ‘femtomolar’ (as in 10^-12)

Line 630 – replace ‘posteriorly’ (which usually refers to anatomical regions located at the back of the body) with ‘ later / afterwards / subsequently’

Author Response

We thank the reviewer for the helpful comments. All changes were implemented, with the exception of the recent reference for the clinical trial. The manuscript focuses on PET tracers for NPY receptors (Y1, Y2 and Y5). The suggested trial uses FDG as a surrogate market for NPY levels so we consider it to be out of scope.

We attach the PRISMA diagram used for the literature review.

Round 2

Reviewer 2 Report

The authors are to be congratulated for addressing all comments proficiently.

Please make sure that the Prisma chart will be part of the paper (as an appendix, or even as a figure) and it won't be lost in the response letter.